# Choice of observation type affects Bayesian calibration of Greenland Ice Sheet model simulations

**Denis Felikson**[1,a], **Sophie Nowicki**[3], **Isabel Nias**[3], **Beata Csatho**[2], **Anton Schenk**[2], **Michael J. Croteau**[4], **and Bryant Loomis**[4]

[1]Cryospheric Sciences Laboratory, NASA Goddard Space Flight Center, Greenbelt, MD, USA
[2]Department of Geology, University at Buffalo, Buffalo, NY, USA
[3]School of Environmental Sciences, University of Liverpool, Liverpool, UK
[4]Geodesy and Geophysics Laboratory, NASA Goddard Space Flight Center, Greenbelt, MD, USA
[a]formerly at: Goddard Earth Sciences Technology and Research Studies and Investigations II, Morgan State University, Baltimore, MD, USA

**Correspondence:** Denis Felikson (denis.felikson@nasa.gov)

**Abstract.** Determining reliable probability distributions for ice sheet mass change over the coming century is critical to refining uncertainties in sea-level rise projections. Bayesian calibration, a method for constraining projection uncertainty using observations, has been previously applied to ice sheet projections but the impact of the chosen observation type on the calibrated posterior probability distributions has not been quantified. Here, we perform three separate Bayesian calibrations to constrain uncertainty in Greenland Ice Sheet (GrIS) simulations of the committed mass loss in 2100 under the current climate, using observations of velocity change, dynamic ice thickness change, and mass change. Comparing the posterior probability distributions shows that the median ice sheet mass change can differ by 119 % for the particular model ensemble that we used, depending on the observation type used in the calibration. More importantly for risk-averse sea-level planning, posterior probabilities of high-end mass change scenarios are highly sensitive to the observation selected for calibration. Furthermore, we show that using mass change observations alone may result in model simulations that overestimate flow acceleration and underestimate dynamic thinning around the margin of the ice sheet. Finally, we look ahead and present ideas for ways to improve Bayesian calibration of ice sheet projections.

## 1 Introduction

Probabilistic sea-level rise projections are critical for coastal decision-making. The Sixth Assessment Report (AR6) from the Intergovernmental Panel on Climate Change has compiled probabilistic projections, with the contribution from the Greenland and Antarctic ice sheets being quantified, for the first time, by higher-order numerical ice sheet models (Fox-Kemper et al., 2021; Nowicki et al., 2020; Goelzer et al., 2020; Seroussi et al., 2020). However, the probabilistic ice sheet projections put forth in AR6 are not conditioned on observations. Performing calibration of projection ensembles using observations allows for refinement of the uncertainties estimated from the ensemble. Additionally, obtaining appropriate probabilities of high-end scenarios is critical for risk-averse coastal planning (Hinkel et al., 2015). For these reasons, it is important to understand the impact that any calibration approach has on the resulting central estimates of the projections as well as on the uncertainties of high-end sea-level rise scenarios.

One method for conditioning ice sheet projection probabilities on observations is Bayesian calibration (Kennedy and O'Hagan, 2001). This approach assigns a score to each model simulation within an ensemble, based on model–observation residuals. The scores are used to construct a likelihood, which quantifies the plausibility of each simulation given the observed data. The likelihood is used to update the prior probability distributions of model parameters and

model outputs (e.g., ice sheet mass change). In this Bayesian framework, the "calibrated" probability distribution is also referred to as the posterior probability distribution. Past studies have used this method to calibrate the relationship between surface mass balance (SMB) and ice sheet surface elevation change (Edwards et al., 2014). Bayesian calibration has been used to constrain Antarctic Ice Sheet (AIS) mass change projections using satellite observations (Ritz et al., 2015) and using paleo-observations (Gilford et al., 2020). The Bayesian calibration framework has been used to explore specific processes, such as the marine ice cliff instability (MICI), in terms of the impact of excluding MICI from AIS projections (Ruckert et al., 2017), and identifying the observational constraints that are needed to assess the likelihood that MICI is occurring (Edwards et al., 2019). The aforementioned studies used observations of ice sheet mass change for calibration; however other observations have also been used. Surface elevation change rates were used to calibrate regional projections of mass change and grounding line retreat in Antarctica (Nias et al., 2019). There has also been recent work that has used a two-step calibration approach, first using observed ice sheet surface velocity and second using reconciled ice-sheet-wide mass change observations from the Ice Sheet Mass Balance Intercomparison Experiment (The IMBIE team, 2018), to calibrate Greenland Ice Sheet (GrIS) mass change projections (Aschwanden and Brinkerhoff, 2022). The use of mass change observations to calibrate ice sheet model ensembles has been proposed as the logical path forward for creating credible sea-level change projections (Aschwanden et al., 2021). However, several different types of observations can be used to calibrate ice sheet projections and there has not yet been a study to systematically quantify the impact of the choice of observation type on the results of the calibration.

In this study, we explore the effect of different observation types on Bayesian calibrations by using velocity change, dynamic ice thickness change, and mass change observations to perform separate calibrations of a Greenland Ice Sheet (GrIS) ensemble, which simulates Greenland's committed contribution to sea-level change over the current century, from 2015 to 2100. Our study builds upon the work of Nias et al. (2023), which presents Bayesian calibration of this ensemble using mass change observations. Here, we use the same calibration approach but perform additional calibrations using observed velocity and thickness changes and compare the results from all three calibrations. Our focus in this study is on how the choice of observation type affects the posterior probability distributions of (1) ice sheet mass change and (2) model parameters and forcings that result from the calibration. We make an effort to be as consistent as possible in the setup of each of the three calibrations, in order to achieve a straightforward comparison. We describe the ice sheet model ensemble in Sect. 2.1 and the observations in Sect. 2.2. Our procedure for the Bayesian calibration is described in Sect. 3. We present results in Sect. 4 and a discussion in Sect. 5.

## 2  Data

### 2.1  Ice sheet model ensemble

We use an ensemble of model simulations that project the committed contribution from the GrIS to sea level over the coming century. The committed contribution captures the mass change of the GrIS in response to current atmospheric and oceanic forcings, independent of future atmospheric or oceanic warming. The committed response can be thought of as the change that is already locked into the ice sheet, which will play out over the coming century. We summarize the model ensemble here and details can be found in Nias et al. (2023). The ice sheet simulations are performed using the Ice-sheet and Sea-level System Model (ISSM; Larour et al., 2012). The simulations are initialized to the year 2007 by inverting for a basal friction coefficient field using surface topography from the Greenland Ice Mapping Project (GIMP; Howat et al., 2014), bed topography from BedMachine v3 (Morlighem et al., 2017a; Morlighem et al., 2017b), and surface velocity from Interferometric Synthetic Aperture Radar (InSAR) satellite data (Joughin et al., 2015b), following the methods of Morlighem et al. (2010). To reduce spurious thickness changes at the start of the simulation, a 30-year relaxation is performed using the 1960–1989 mean surface mass balance (SMB) from the Regional Atmospheric Climate Model (RACMO2.3p2; Noël et al., 2019) and keeping the ice extent fixed to the initial state. From this relaxed initial state in 2007, an ensemble of 137 forward runs is performed for the time period 2007–2100 by using Latin hypercube sampling (McKay et al., 1979; Eglajs and Audze, 1977) from a multidimensional uniform distribution of uncertainty, which defines the prior distributions of basal friction, ice viscosity, and surface mass balance. Basal friction is varied by applying a spatially constant factor sampled from a uniform distribution with bounds of $\pm 50\%$ to the field obtained from the inversion procedure, and this field is kept fixed through time. Ice viscosity is varied by applying a spatially constant offset sampled from a uniform distribution with bounds of $\pm 10\,\mathrm{K}$ to the initial temperature field, which is then converted to ice viscosity (Cuffey and Paterson, 2010), and this field is kept fixed through time. The SMB field is varied by (1) adding an offset and (2) changing the seasonal amplitude of the mean 2001–2015 SMB from RACMO2.3p2 (Noël et al., 2019). The SMB offset is varied by applying a spatially constant factor sampled from a uniform distribution with bounds of $\pm 30\%$, and the seasonal amplitude is varied by applying a spatially constant factor sampled from a uniform distribution with bounds of 0 to 2, where a factor of 2 represents a doubling of the seasonal amplitude of SMB, and a factor of 0 represents elimination of the seasonal amplitude. The variations in SMB offset and seasonal amplitude are applied spatially uniformly across the entire ice sheet, and the annual SMB pattern is repeated yearly. From 2007 to 2015, the ice front in the model is specified using observa-

tions of outlet glacier terminus positions (Moon and Joughin, 2008; Joughin et al., 2015a) via a level-set method (Bondzio et al., 2016). From 2015 onward, the terminus positions are held fixed in their 2015 locations, allowing the ice sheet to adjust to the 2007–2015 terminus perturbations and, thus, yielding the committed contribution from the GrIS, independent of future terminus position change. While the relaxation removes initial spurious model behavior, some model drift inevitably remains at the start of the simulation. The goal of the Bayesian calibration approach is to calibrate the projections such that the true drift of the GrIS over 2007–2015 is captured by the ensemble. However, initial model drift is caused both by the true imbalance of the GrIS in 2007 and by erroneous model inconsistencies, and these two contributions are not separated. To provide examples of simulated model states, we show modeled velocities and surface elevations from the lowest-weighted (Fig. S1) and highest-weighted (Fig. S2) ensemble members from the mass change calibration (Nias et al., 2023).

## 2.2 Observations

### 2.2.1 Velocity change

The first observation type that we use for calibration is velocity change. Velocity change observations are computed using the Making Earth System Data Records for Use in Research Environments (MEaSUREs) Greenland Ice Sheet Velocity Map from Interferometric Synthetic Aperture Radar (InSAR) Data, Version 2 (Joughin et al., 2015b). This dataset uses several sources of InSAR measurements to compile winter ice-sheet-wide velocity maps on an annual basis. We use the maps from 2007 and 2015 to obtain velocity change over these years. We calculate velocity change as the difference between the 2015 and 2007 velocity maps, and uncertainty is calculated as the root sum of squares of the two associated uncertainty maps.

### 2.2.2 Dynamic ice thickness change

The second observation type that we use for calibration is dynamic ice thickness change. This quantity is a useful measure of how out of balance the ice dynamics are with the climate, and, thus, it is a good metric for evaluating the ensemble. Ice sheet surface elevation change time series are obtained from airborne and spaceborne laser altimetry using the Surface Elevation Reconstruction and Change (SERAC) method (Schenk and Csatho, 2012; Shekhar et al., 2020). To obtain dynamic surface elevation change, we account for thickness change anomalies due to both surface and firn processes by applying the Institute for Marine and Atmospheric research Utrecht (IMAU) Firn Densification Model (FDM), which simulates thickness change of the firn, forced by RACMO2.3p2 (Ligtenberg et al., 2018), which simulates thickness changes due to SMB. We subtract the firn thickness

change anomalies from the SERAC surface elevation change, with anomalies referenced to the average over the time period 1960–1979 (Fig. S3). We then fit a continuous function to the discrete SERAC estimates through time. We assume that the dynamic surface elevation change from SERAC is equivalent to the dynamic ice thickness change because glacial isostatic adjustment rates around the margin of the GrIS are orders of magnitude smaller than the dynamic surface elevation change rate signal (Wake et al., 2016). Time series with a magnitude of dynamic ice thickness change greater than 5 m over the entire SERAC time series are typically characterized by complex temporal behavior; at these locations, we use the Approximation by Localized Penalized Spline (ALPS) method (Shekhar et al., 2020) to approximate a continuous function through time. Time series with a magnitude of dynamic ice thickness change less than 5 m over the entire SERAC time series exhibit less complex behavior, and we fit a cubic polynomial to the discrete SERAC estimates at these locations. Our dynamic ice thickness change estimates represent the thickness change of the ice that are caused by changes in flux divergence at each SERAC measurement location and time.

We sum the annual dynamic ice thickness change over the time period 2007–2015, only considering those SERAC locations that have an estimate for each year between 2007 and 2015. This results in a set of more than 16 000 data points over the ice sheet in irregularly distributed locations, with higher density around the ice sheet margin where airborne altimetry provides increased spatial sampling beyond the spaceborne altimetry (Csatho et al., 2014).

To assign uncertainties, we combine the approximation errors of ice thickness change from the ALPS and polynomial fits and inflate them to account for errors in the FDM and SMB estimates. We investigate fit errors from ALPS at locations where the magnitude of dynamic ice thickness change is $> 5$ m and find errors to be $\sim 1$ m over the calibration time span. Similarly, fit errors from the cubic polynomial at locations where the magnitude of dynamic ice thickness change is $< 5$ m are $\sim 0.1$ m. We conservatively inflate these to account for uncertainty in the SMB and FDM and use 1.4 and 0.14 m as the uncertainties in dynamic ice thickness change where dynamic ice thickness change is $> 5$ and $< 5$ m, respectively. This achieves an expected spatial pattern in uncertainties, with larger uncertainties around the GrIS margin, where dynamic ice thickness change is relatively large, and smaller uncertainties in the GrIS interior, where dynamic ice thickness change is relatively small (Fig. 1).

### 2.2.3 Mass change

The third observation type we use for calibration is mass change observations. Mass change observations of the GrIS are derived from the high-spatial-resolution NASA Goddard Space Flight Center (GSFC) global mascon trend solution (release 06), obtained from the Gravity Recovery and

Climate Experiment (GRACE) and the Gravity Recovery and Climate Experiment Follow-On (GRACE-FO) (Loomis et al., 2021). These observations provide mass change trends globally within 1 arcdeg equal area cells. Mass change over the calibration time span of 2007–2015 is calculated from the observed trends per mascon over the GrIS. Mass change uncertainties are calculated from statistics of the differences between the GSFC high-spatial-resolution mascon trend solution (Loomis et al., 2019) and the Gravity Observation Combination release 06 (GOCO-06) spherical harmonic model (Kvas et al., 2021). The mass change uncertainties were assessed separately over the margin and the interior of the GrIS, and the resulting uncertainties are 4 cm w.e. yr$^{-1}$ for the mascons around the GrIS margin and 1 cm w.e. yr$^{-1}$ for the mascons in the GrIS interior (Fig. S4). This results in a spatial pattern of mass change uncertainties that is similar to that for dynamic ice thickness change, with relatively higher uncertainties around the GrIS margin and relatively lower uncertainties in the GrIS interior.

## 3    Methods

### 3.1    Observation preprocessing

In our Bayesian calibration approach (described in Sect. 3.3), we assume that the model–observation residuals are independent from one another. In other words, we assume no spatial correlation between residuals. In order to use this assumption for velocity and thickness change, we coarsen the input observation datasets by establishing a regular polar stereographic grid with 50 km by 50 km grid cells and calculating the mean observed quantity and uncertainty in each grid cell. Additionally, we discard gridded velocity change observations for grid cells with less than 75 % coverage. We transform the coordinates of the dynamic ice thickness change time series locations from their native projection of Universal Transverse Mercator (UTM) zone 24N (EPSG coordinate system 32624) to a polar stereographic north projection (EPSG coordinate system 3413). The mass change observations are aggregated within drainage basins using the delineations described in Rignot et al. (2011). Calculating model–observation residuals at the drainage basin scale helps to ensure that they are independent (Nias et al., 2023). Gridded observations and their uncertainties are shown in Fig. 1. Note that an improved method for computing basin-scale uncertainties for observed mass change is presented in Loomis et al. (2021), and uncertainties calculated with this method are slightly lower than the ones used in Nias et al. (2023) (Fig. S4).

### 3.2    Model output preprocessing

To calculate model–observation residuals, we must calculate model quantities that match the observed quantities. Velocity is a model state variable and we can compare modeled velocity change directly with observations. Modeled dynamic ice thickness change is calculated by subtracting the SMB anomaly (in units of ice thickness equivalent) from the modeled ice thickness change over 2007–2015. SMB anomaly is calculated by subtracting the mean 1960–1979 SMB from the prescribed SMB forcing for each simulation in the ensemble. The prescribed SMB forcing for the model simulations is obtained from RACMO2.3p2 (Noël et al., 2019) and the mean 1960–1979 SMB is calculated from this same model dataset. Our calculation of modeled dynamic ice thickness change corresponds to the method used to calculate observed dynamic ice thickness change, which uses an FDM to estimate the thickness change associated with surface and firn processes and removes this from the observed surface elevation change to obtain dynamic ice thickness change. Modeled mass change is calculated by multiplying the modeled volume change, which is also a model state variable, by a constant ice density (917 kg m$^{-3}$) to obtain mass change. Modeled mass change is defined at the resolution of model elements and this is then converted to GRACE mascon space using the GRACE resolution operator. Details on this conversion can be found in Nias et al. (2023), and the resolution operator is described in Loomis et al. (2019). Note that the mass loss due to the imposed ice front retreat is not included in the modeled mascon equivalents because it is computed from ice thickness change at model nodes where ice is present throughout the calibration time period (2007–2015). However, this unaccounted for mass loss in the modeled mascons is negligible compared to the basin-scale ice sheet mass change over this time period used in the calibration.

The modeled quantities are regridded spatially to match the observational quantities. Mean modeled velocity change and dynamic ice thickness change are calculated within the grid cells defined by the same 50 km by 50 km grid used for the observational datasets. Modeled mass change is aggregated within the same drainage basins as used for aggregating the observed mass change (Nias et al., 2023).

### 3.3    Bayesian calibration

We use a Bayesian calibration approach to refine the spread in simulated GrIS mass change and the uncertainty in model parameters and forcings from our ice sheet model ensemble. A thorough treatment of this approach can be found in Kennedy and O'Hagan (2001), and we summarize the relevant steps here. In this approach, we calibrate the prior probability distributions of (1) ice sheet mass change and (2) model parameters and forcings using observations of the ice sheet to obtain posterior probability distributions of these quantities. Bayes' theorem states that

$$p(\boldsymbol{m}|\boldsymbol{d}) \propto p(\boldsymbol{m})\,p(\boldsymbol{d}|\boldsymbol{m}), \tag{1}$$

where $p(\boldsymbol{m})$ is the prior probability distribution of model parameters, forcings, or simulated outputs (e.g., mass change); $p(\boldsymbol{d}|\boldsymbol{m})$ is the likelihood function of $\boldsymbol{m}$ given observations;

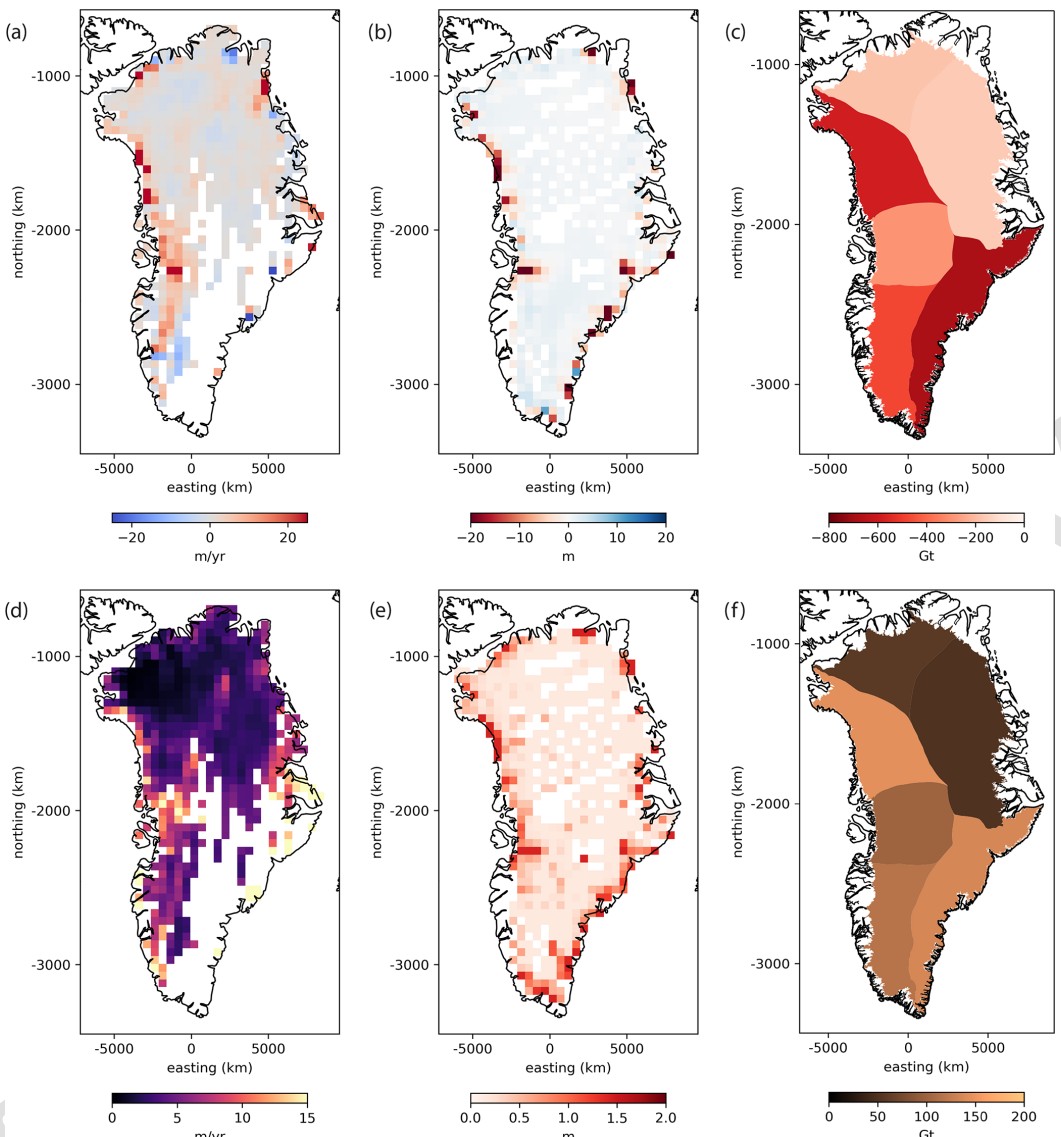

**Figure 1.** Gridded observations and their uncertainties for the 2007–2015 calibration time period. Panels **(a)**–**(c)** show velocity change **(a)** and dynamic ice thickness change **(b)**, which are gridded on a regular polar stereographic grid with 50 km by 50 km grid cells, and mass change **(c)**, which is aggregated by basin (Rignot et al., 2011). Panels **(d)**–**(f)** show velocity change uncertainty **(d)**, dynamic ice thickness change uncertainty **(e)**, and mass change uncertainty **(f)**, with the same gridding as the observations in the top row (**a**, **b**, and **c**). TS1

and $\boldsymbol{d}$ and $p(\boldsymbol{m}|\boldsymbol{d})$ are the posterior probability distribution of $\boldsymbol{m}$.

To construct the likelihood, we use model–observation residuals to assign a likelihood score for each ensemble member. We assume that the residuals are independent, are identically distributed, and follow a normal (Gaussian) distribution. Under these assumptions, the score for the $j$th ensemble member is calculated as

$$s_j = \exp\left[-\frac{1}{2}\sum_i \frac{(f_i^j - z_i^j)^2}{(\sigma_i)^2}\right], \qquad (2)$$

where $f$ is the modeled quantity, $z$ is the observed quantity, $\sigma^2$ is the variance of the residual, and $i$ is a spatial index. Note that the multiplicative constant that typically appears in the equation for a normal distribution has been discarded because of the normalization of the scores, $s_j$, that is done later. For the velocity and thickness change calibrations, the spatial index, $i$, represents each 50 km by 50 km grid cell within which observations are defined, and for the mass change calibration, the spatial index represents each basin within which mass change observations are aggregated. Uncertainty in the residual includes both the observational uncertainty, $\sigma_{o,i}$, and the model (also called structural) uncertainty, $\sigma_{m,i}$, both of

**Table 1.** Values for the multiplier $k$ used to calculate structural model uncertainty from observational uncertainty.

| Observation type | $k$ |
| --- | --- |
| Velocity change | 75 |
| Thickness change | 150 |
| Mass change | 8 |

which vary in space as indicated by the spatial index $i$:

$$\sigma_i = \sqrt{\sigma_{o,i}^2 + \sigma_{m,i}^2}. \tag{3}$$

The observational uncertainty for each observation type, $\sigma_{o,i}$, is shown in Fig. 1. Values for $\sigma_{m,i}$ are typically specified in an ad hoc manner as a multiple of $\sigma_{o,i}$ (e.g., Edwards et al., 2014; Nias et al., 2019). The underlying assumption of this approach is that the accuracy of the model is less than the accuracy of the observations (Edwards et al., 2019). Here, we set $\sigma_{m,i} = k\sigma_{o,i}$ and we manually adjust $k$ for each observation type such that the peaks of the posterior probability distributions of GrIS committed contribution to global mean sea level (GMSL) in 2100 from our three calibrations are approximately equal (Fig. 2). The selected value of $k$ will also affect the width of the posterior probability distribution, with smaller values resulting in narrower distributions (Figs. S5 and S6). This choice allows us to compare the calibration results in a straightforward manner. The multiplier, $k$, varies for each observation type and the values that we use are shown in Table 1.

Once the scores, $s_j$, are calculated for each calibration, they are normalized such that $\sum_{j=1}^{n} s_j = 1$, where $n$ is the total number of ensemble members, resulting in weights that are applied to the prior distributions of (1) mass change and (2) model parameters and forcings to obtain the posterior distributions. The weights assigned to each ensemble member are used to estimate the posterior probability density function using Gaussian kernel density estimation (Scott, 1992).

We perform three separate calibrations using observations of (1) velocity, (2) thickness, or (3) mass change and compare the posterior probability distributions of GrIS committed contribution to GMSL in 2100, as well as model parameters and forcings obtained from all three calibrations.

## 4 Results

The calibrated posterior probability distributions of GrIS committed contribution to GMSL and the associated statistics differ between the three calibrations (Fig. 2, Table 2). The median GMSL varies significantly, based on the type of observation chosen for calibration. The median GMSL from the thickness change calibration is $-6.4\,\text{mm}$, whereas the median GMSL from the mass change calibration is $33.8\,\text{mm}$. As shown by the 95th percentile of the posterior distribution,

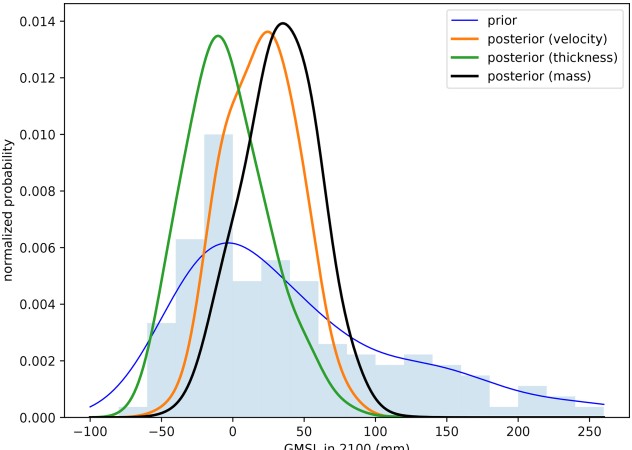

**Figure 2.** Posterior probability distributions of Greenland's committed contribution to global mean sea level (GMSL) in 2100 from three Bayesian calibrations: velocity (orange), thickness (green), and mass (black) change observations. Prior distribution is shown as a blue curve and the histogram of the prior population is shown as a blue bar graph. Note that the $y$ axes are normalized probabilities, such that the integrals under the prior histograms and the posterior probability curves equal 1.

all three calibrations decrease the probability of the highest possible contributions to GMSL; the prior probability distribution of GMSL has a 95th percentile of 213.3 mm and the calibrations decrease this to $\leq 77.8\,\text{mm}$ (Table 2). The 5th percentiles are all negative, indicating that all three calibrations result in a possibility that Greenland will accumulate mass, decreasing GMSL. Like the 95th percentiles, the 5th percentiles also decrease in magnitude as a result of the calibration from $-60.3$ down to $-14.7\,\text{mm}$ for the mass change calibration. In other words, all three calibrations serve to narrow the probability distribution of the committed contribution to GMSL from the GrIS. The choice of observation type also has a strong effect on the cumulative probability of high-end scenarios (GMSL $> 50\,\text{mm}$). Calibration using thickness change observations yields a 14.2 % probability of GMSL $> 50\,\text{mm}$, whereas calibration using mass change observations yields a 28.1 % probability of GMSL $> 50\,\text{mm}$, a 2-fold difference in probabilities. All three calibrations reduce the probability for the most extreme scenario (GMSL $> 100\,\text{mm}$) from 23 % to $\leq 0.6\%$.

Posterior probability distributions of model parameters and forcings for all three calibrations are shown in Fig. 3. In all cases, the extreme values of all parameters and forcings are de-weighted, and the peaks of the posterior distributions occur near the central values. The posterior distributions we obtain from the three calibrations are similar for ice temperature and SMB seasonal amplitude. However, for the basal friction multiplier, there are notable differences in the posterior probability distributions between the three calibrations. Calibration using thickness change observations re-

**Table 2.** Statistics for GMSL median (mm), 5th and 95th percentiles (mm), and probability of GMSL larger than 50 and 100 mm (**P**(GMSL >50 mm) and **P**(GMSL > 100 mm), respectively).

| | Median | Percentiles | | **P**(GMSL > 50 mm) | **P**(GMSL > 100 mm) |
|---|---|---|---|---|---|
| | | 5th | 95th | | |
| Prior | 26.6 | −60.3 | 213.3 | 38.8 % | 23.0 % |
| Posterior (velocity calibration) | 20.4 | −23.9 | 64.5 | 14.2 % | 0.1 % |
| Posterior (thickness calibration) | −6.4 | −51.0 | 51.2 | 5.3 % | 0.1 % |
| Posterior (mass calibration) | 33.8 | −14.7 | 77.8 | 28.1 % | 0.6 % |

sults in a posterior probability distribution that is skewed towards higher values of the basal friction multiplier, whereas the velocity change and mass change calibrations result in posterior probability distributions that are more symmetric. Similarly, for SMB mean shift, the thickness change calibration results in higher probabilities of positive shifts than the velocity and mass change calibrations.

We find that three different ensemble members are scored highest across the three calibrations, with notable differences in the spatial patterns of their model–observation residuals (Fig. 4). All three calibrations result in larger residuals around the margin than in the interior of the ice sheet, an expected result due to the fact that the magnitude of change is largest around the margin. Residuals for the same observational quantity used in each calibration are smaller than residuals from the other calibrations (Fig. 4a, e, i). In other words, velocity change residuals are smallest for the highest-weighted ensemble member from the velocity calibration and similarly for the thickness and mass change calibrations. Looking at velocity change residuals, all calibrations overestimate velocity change along the eastern margin of the ice sheet and underestimate velocity change along the northern margin (Fig. 4a–c). Along the western margin, the residuals differ across the three calibrations, with the highest-weighted ensemble member from the thickness calibration showing a large underestimate of velocity change (Fig. 4b). For outlet glaciers that have accelerated during the 2007–2015 calibration time period, this means that the acceleration is underestimated by the thickness change calibration. The velocity change and mass change calibrations yield highest-weighted ensemble members with a slight underestimate and slight overestimate of velocity change along the western margin, respectively (Fig. 4a and c). The thickness change residuals are similar in their spatial structure across all three calibrations (Fig. 4d–f). Dynamic ice thickness change residuals is positive around almost the entire margin of the ice sheet, meaning that the model underestimates the magnitude of observed dynamic thinning. The exception is the highest-weighted ensemble member from the thickness change calibration, which overestimates thinning around Jakobshavn Isbræ. The mass change residuals have similar features across the three calibrations (Fig. 4g–i). Mass change residuals are positive in the east, southwest, and north, meaning the model

underestimates the magnitude of observed mass loss in those regions. In the northeast, mass change residuals are negative, meaning the model overestimates the magnitude of observed mass loss there. In the northwest, there is a mix of positive and negative mass change residuals. Using the residuals' root sum of squares (RSS) as a measure, the velocity change residuals show the highest sensitivity to the calibration choice, with a 38 % difference in residuals' RSS between the velocity and thickness change calibrations. The thickness change and mass change residuals have less sensitivity, with a 16 % difference in thickness change residuals' RSS between the velocity and thickness calibrations and an 18 % TS2 difference in mass change residuals' RSS between the mass and thickness calibrations.

## 5 Discussion

The choice of observation type strongly affects the results of the calibration. This choice affects the posterior probability distributions of GrIS committed contribution to GMSL in 2100 (Fig. 2), as well as the posterior probability distributions of model parameters and forcings (Fig. 3). The median estimates of GMSL in 2100 can differ by 119 % (Table 2) and, more importantly, the probabilities of "high-end" scenarios can be very different, depending on the observation type used in the calibration. The posterior probability of > 50 mm of committed GMSL in 2100 is a factor of 2 larger for the calibration using mass change than the calibration using thickness change observations (Table 2). The "highest-end" scenarios, however, are not sensitive to the choice of observation; all three calibrations effectively rule out the possibility of > 100 mm of GMSL, regardless of the observation type, essentially eliminating the 23 % cumulative probability of GMSL > 100 mm in the prior distribution. The lower probability of GMSL > 50 mm in the thickness change calibration is caused by the thickness change calibration assigning higher weights to ensemble members with higher basal friction coefficients and a higher SMB mean shift (Fig. 3a and c, respectively).

Basal friction is the most sensitive parameter to choice in observation type (Fig. 3a). The velocity change calibration yields a posterior distribution that has a higher peak and nar-

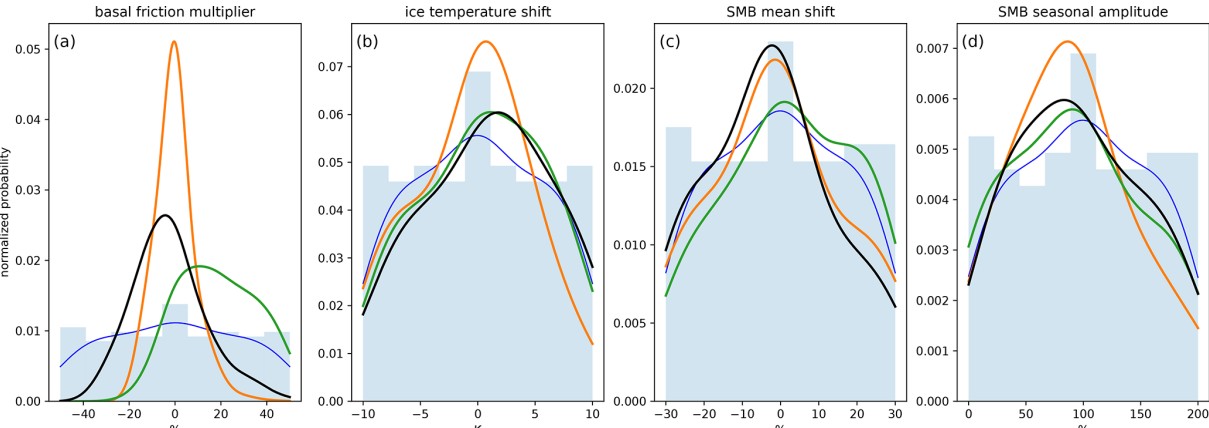

**Figure 3.** Prior and posterior probability distributions of model parameters and forcings: basal friction multiplier **(a)**, ice temperature shift **(b)**, SMB mean shift **(c)**, and SMB seasonal amplitude **(d)**. Prior probability distribution is shown as a histogram (light-blue bars) and as a Gaussian approximation (blue curve). Posterior probabilities are shown for Bayesian calibrations using ice sheet surface velocity (orange), ice sheet thickness change (green), and ice sheet mass change (black). Note that the *y* axes are normalized probabilities, such that the integrals under the prior histograms and the posterior probability curves equal 1.

rower spread for basal friction than the other two calibrations. Posteriors for the other parameters and forcings (ice temperature, mean SMB, and seasonal SMB) are less sensitive to the choice of observation used in the calibration. The narrower posterior distribution of basal friction from the velocity change calibration also corresponds to a slightly narrower posterior probability distribution of GMSL in 2100 from that calibration (Fig. 2), as shown by the percentiles in Table 2.

Calibration using dynamic ice thickness change results in a posterior probability distribution of GrIS committed contribution to GMSL in 2100 that is quite different from the velocity change and mass change calibrations. Although the spread in the posterior probability distributions of GMSL in 2100 is similar across the three calibrations, as seen from the percentiles in Table 2, the median estimate from the thickness change calibration (−6 mm) is negative, indicating that the GrIS is increasing in mass, whereas the median estimate from the velocity change and mass change calibrations is positive (20 and 34 mm, respectively), indicating that the GrIS is decreasing in mass. This corresponds to the thickness change calibration assigning higher likelihood to simulations with higher basal friction and a positive shift in mean SMB than the velocity and mass change calibrations (Fig. 3a and c, respectively). Simulations with higher basal friction result in a slower propagation of mass loss into the interior of the ice sheet in response to terminus retreat (Nias et al., 2023), and this can be seen in the larger negative velocity change residuals that extend further into the ice sheet interior for the highest-weighted ensemble member from the thickness calibration, indicating that ice sheet acceleration is underestimated by the model (Fig. 4b).

Several factors may contribute to discrepancies seen across the three calibrations. For the dynamic ice thickness change calibration, a potential source for a bias is the estimate of firn thickness change that is used in the calculation of observed dynamic ice thickness change. A bias can be caused by a bias in the estimated trend of firn thickness change either over the 1960–1979 baseline time period or over the 2007–2015 calibration time period; this bias will directly translate to a bias in the observed dynamic ice thickness change and can then bias the posterior probability distribution of ice sheet mass change. Additionally, all three calibrations may be impacted by unique issues related to spatial sampling of the observations. Mass change observations from satellite gravimetry can be affected by "signal leakage", meaning that mass change of peripheral glaciers and ice caps proximal to the GrIS may contaminate the mass change observations around the ice sheet margin (Loomis et al., 2021). On the other hand, the ice sheet model may include some of this peripheral ice mass as part of its domain, as there is not a standard way to separate the GrIS from peripheral ice masses (Goelzer et al., 2020). Finally, both velocity change and dynamic ice thickness change observations can be affected by data gaps. This is most notable in the velocity change observations, which are missing for a portion of the southeast margin of the GrIS (Fig. 1a); however the dynamic ice thickness change observations also have some data gaps in the interior of the GrIS (Fig. 1b). This gap in the velocity change observations means that the model simulations are not scored based on their ability to capture velocity change in southeast Greenland. In contrast, the thickness change and mass change observations provide adequate data coverage to sample the southeast of the GrIS. To fill the velocity data gap, a later start year could be selected for the calibration. However, selecting velocity observations from a later year will shorten the calibration time span and, thus, potentially remove information from the likelihood. The trade-off between filling data

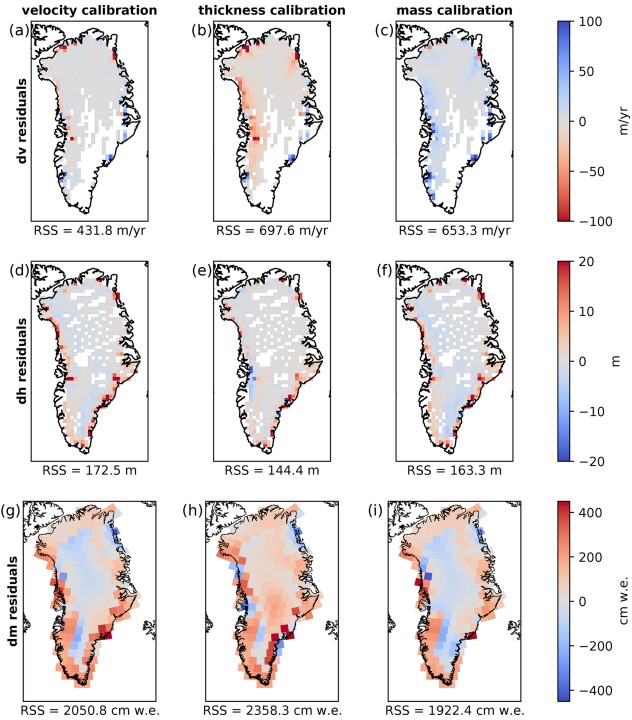

**Figure 4.** Residuals (modeled minus observed) for the highest-weighted ensemble members from Bayesian calibration using velocity change (column 1), thickness change (column 2), and mass change (column 3) observations. Residuals of velocity (row 1), thickness (row 2), and mass (row 3) change over the calibration time period of 2007–2015 are shown. Note that mass change residuals are shown at the resolution of individual mascons, rather than aggregated within basins, in order to provide additional detail. We provide the root sum of squares (RSS) of the residuals shown below each map. TS3

gaps in velocity observations and shortening the calibration time span should be explored in future work.

Differences among the three calibrations in terms of their residuals can provide insight into biases in the model ensemble beyond what can be gleaned from any one of the calibrations alone. For example, the highest-weighted ensemble member from the mass change ensemble overestimates acceleration (Fig. 4c) and underestimates dynamic thinning along almost the entirety of the GrIS margin (Fig. 4f). As discussed above, the dynamic thickness change observations may be affected by errors in the estimate of firn thickness change, but this discrepancy around the margin is not affected by this source of error because firn does not persist around most of the ice sheet margin. Combining information from all available sources of observations will make the calibration more robust to systematic uncertainties in the observations, as discussed in the previous paragraph. Additionally, utilizing multiple observations may reveal the presence of compensating model errors that allow model simulations to correctly reproduce one observation type but for an incor-

rect reason. Aschwanden and Brinkerhoff (2022) have taken a step in this direction, by performing a two-step calibration process of GrIS projections, first using observations of velocity to calibrate a subset of model parameters and second using observations of mass change to calibrate the rest of the parameters. This is a notable advancement in utilizing information from multiple observations for Bayesian calibration, and their use of velocity observations in calibration is somewhat akin to the use of 2007 velocities to initialize our model ensemble (Nias et al., 2023). Adding information from dynamic thickness change observations would build upon their approach and help to further constrain uncertainty in the projections. However, more work needs to be done to better understand the structural model uncertainties for the different observation types in order to combine their likelihoods.

The ad hoc approach used here and elsewhere (Nias et al., 2023; Edwards et al., 2014) to estimate structural model uncertainty as a multiple of the observational uncertainty means that the spatial structure of the observational uncertainty is preserved in the residual uncertainty, $\sigma$. Therefore, it is important to have accurate estimates of the spatial structure of errors provided by observational data products. On the other hand, to move beyond this ad hoc approach, it is necessary for the ice sheet modeling community to develop an improved understanding of the structural model uncertainty in the quantities that are used for calibration.

We tested several values for the value of the multiplier, $k$, and found that it affects the shape of the posterior probability distribution, with smaller values resulting in narrower distributions and lower peaks, but it does not affect the ensemble member that is weighted highest (Figs. S5 and S6). We also investigated eliminating the structural model uncertainty term in the calibration altogether (i.e., by setting $k$ equal to zero). However, this results in calculated weights, $w_j$, that are all equal to zero due to our formulation of the likelihood function as a normal (Gaussian) distribution. Mathematically, this occurs because the numerator in the likelihood equation becomes relatively much larger in magnitude than the denominator. Future work should investigate other functional forms for the likelihood and the impact of the functional form, along with the choice of the value for $k$, on the results.

For our study, we used an ensemble of GrIS committed contribution to GMSL, which quantifies the changes that are locked in to the ice sheet, independent of any additional future atmospheric or oceanic forcing. Greenland's commitment results in a contribution to GMSL that is similar in magnitude to the contribution due to future climate anomalies under the Representative Concentration Pathway (RCP2.6) and Shared Socioeconomic Pathway (SSP1-26) scenarios (Edwards et al., 2021) but lower in magnitude than the contributions under the higher emissions scenarios. Additional work is needed to perform similar Bayesian calibrations of GrIS ensembles forced with future climate anomaly projections (e.g., Goelzer et al., 2020; Aschwanden et al., 2019). How-

ever, we hypothesize that there will be significant discrepancies among the posterior distributions of GrIS contribution to GMSL in these ensembles, although the relative differences between calibrations may be smaller in an ensemble of forced mass loss under the higher emissions scenarios than the relative differences for our committed mass loss ensemble, due to the fact that the magnitudes of ice sheet mass change will be larger in projections forced with higher emissions scenarios.

## 6 Conclusions

Our study presents three calibrations using three different observation types (velocity, thickness, and mass change) of the GrIS over an 8-year time period. The choice of observation type leads to important differences in the posterior probability distributions of GrIS committed contribution to GMSL in 2100. It has been proposed that mass change observations should be used to calibrate ice sheet model projections (Aschwanden et al., 2021). However, as we have shown, calibrating with any one observation type will not necessarily result in high-weighted ensemble members that will reproduce certain desired behaviors, such as reproducing observed outlet glacier dynamic thinning or acceleration. More work must be done to better understand the impact of various choices made during the calibration process and to develop better approaches to incorporating information from different observation types.

Using Bayesian calibration to constrain uncertainty in ice sheet ensembles still has many open questions. We have clearly shown how Bayesian calibration can refine uncertainties in ice sheet projections but future work should explore additional choices, such as the method for specifying model structural uncertainty, the time span over which the calibration is done, the use of time series of observations rather than a snapshot of change, and the use of additional metrics derived from these observations. Additionally, future work can move away from the simplifying assumption that we have made that model–calibration residuals are uncorrelated and, instead, quantify the correlation and incorporate that through a covariance matrix into the calibration procedure. Finally, the modeling community should develop robust methods to quantify structural model uncertainty for velocity change, dynamic ice thickness change, and mass change, which could then be used to perform a multivariate calibration using all three observation types simultaneously. Ultimately, the goal is to make use of all of the observation types to get the best possible calibration, although, as we have shown, utilizing different observation types in separate calibrations can yield additional insight into biases in the model ensemble.

*Code and data availability.* Velocity observations from the Making Earth System Data Records for Use in Re-

search Environments (MEaSUREs) project are available at https://doi.org/10.5067/OC7B04ZM9G6Q (Joughin et al., 2015b). Mass change observations derived from the Gravity Recovery And Climate Experiment (GRACE) and Gravity Recovery And Climate Experiment Follow-On (GRACE-FO) are available at https://doi.org/10.5281/zenodo.10037961 TS4 (Loomis et al., 2023; Loomis et al., 2021). Code for Bayesian calibration is available at https://doi.org/10.5281/zenodo.10038131 TS5 (Felikson, 2023). Surface elevation estimates from the Surface Elevation Reconstruction and Change (SERAC) approach are available at https://doi.org/10.5281/zenodo.7324429 (Csatho et al., 2022). TS6

*Supplement.* The supplement related to this article is available online at: https://doi.org/10.5194/tc-17-1-2023-supplement.

*Author contributions.* DF and SN conceptualized the study. IN designed the model ensemble and performed the mass change calibration. DF performed the velocity change and thickness change calibrations and carried out the comparison of the results from the three calibrations. BC and AS prepared the thickness change observations. MC and BL prepared the mass change observations. DF prepared the paper with contributions from all co-authors.

*Competing interests.* The contact author has declared that none of the authors has any competing interests.

ther geographical representation in this paper. While Copernicus Publications makes every effort to include appropriate place names, the final responsibility lies with the authors.

*Financial support.* This research has been supported by the National Aeronautics and Space Administration (grant nos. NNH19ZDA001N-SLCST, 80NSSC21K1734, NN-H19ZDA001N-GRACEFO, 80NSSC17K0611, and 80NSSC21K0915).

*Review statement.* This paper was edited by Johannes J. Fürst and reviewed by two anonymous referees.

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
