# Peer review of "Choice of observation type affects Bayesian calibration of Greenland Ice Sheet model simulations"

_EGUsphere, 2022_

## Referee Comment (RC1)

**Review Felikson et al. (2022)**

This study aims to constrain and improve projections of the Greenland ice sheet evolution by making use of changes in three different observed quantities (mass changes, velocity changes and dynamic thickness changes) that are a measure the Greenland ice sheet dynamics. So far, Bayesian calibration of Greenland ice sheet model projections have been based on a single observation type.

I think the study presents a very useful framework to improve ice sheet model projections based on observations. However, as also stated in the study, using three different observation types do not give a clear answer on what is the best observation type to constrain the ice sheet model behavior and there might be a better metric when different observation types are combined. Therefore I am wondering whether the study as presented in the present form is convincing enough that model projections have been improved by using Bayesian calibration using different observation types. In addition, I would like to propose some minor clarifications on the methodological choices that you can read below.

**Main comments**

You explain that you coarsen the resolution in order to remove the spatial correlation between the different observations. However, if you look at Figure 1, there is still a strong correlation between the observed velocity change and the dynamic thickness change, especially for the fast outlet glaciers. It looks like it is a workaround to justify the neglection of a covariance matrix. I am also wondering if you are not losing too much important, detailed information by coarsening the resolution only to justify that the different variables are not correlated. Especially velocity changes at 50 km resolution remove a lot of detail. Did you investigate the posterior probability distribution for higher spatial resolutions?

The model is initialized to the year 2007 and observations are used until 2015. What is the influence of the length of the observation time on the results? Could it be useful to use a longer observational time series, for instance by initializing the model to the year 2000 to increase the observational length?

You refer in the methodology to the Nias et al. paper in review, but there is no preprint available (or no reference in the reference list) so it is impossible to acquire the right information. For instance, the model ensemble details can be found in Nias et al. (in review). It would be interesting to have them in this manuscript as well. Also, you mention on L153 that the modeled mass change is aggregated within the same drainage basin as used for aggregating the observed mass change. Can you explain what that means, why and how that is done?

The goal is to narrow the committed Greenland ice sheet change projections. However, you are changing not only model parameters but also the SMB, which you define as a forcing. Can you still speak of committed sea-level change if you change the SMB forcing?

It would be nice to add some figures about the modelled Greenland ice sheet state. You show only 4 figures that basically focus only on the posterior probability distribution of committed GMSL rise due to melting of the Greenland ice sheet.

**Minor comments**

L61: It could be useful to give the values for the basal friction factor, the ice viscosity offset and the SMB offset for the 137 forward runs in the supplementary information. Could you add a justification for the offsets chosen?

L151: The modeled quantities are regridded to 50 km x50 km. What is the original resolution of the model results?

L173-L175: The model uncertainty is a linear function of the observational uncertainty and you assign different multiplication factors for the different calibration to match the peak in the posterior distribution. Could you elaborate a bit more on the choice of these values? Because it looks contradictory to what you say on L185, that the median and maximum a posteriori GMSL are far apart.

L207: That sounds like a logical consequence of the larger changes along the margin of the ice sheet.

L252: You discuss the firn thickness change as a potential bias. What is the modelled firn thickness change? Could you show a figure? Also you report a potential bias of 10 cm per year in the ice sheet interior. What is this number based on?

**Typo's**

L124: and and

L274: Open questions?

L284: correct 'to on the choice'

**Figures**

Figure 1: Something went wrong with the labels: please add/modify the labels (a), (b), (c), (d), (e) and (f). Please also adapt the colormap labels to make them uniform with the indication of the variable and the units between brackets. I would also add the resolution for the different gridded observations.

Figure 3: I do not see the blue curve with the Gaussian approximation.

---

## Author Response (AR1)

Author response with paper revision submission

Dear Editor and Reviewers of The Cryosphere,

We would like to again thank the editor and reviewers for the constructive comments and suggestions, which have helped us improve the clarity and strengthen the arguments in our paper. We are submitting our revised manuscript having addressed all comments below. While making our revisions, there were a few cases where we deviated slightly from our previous response to reviewers and we document these cases below with an additional author response shown in blue text, below our original author response. For all other author responses where we do not provide any additional clarifications, we have addressed those reviewer comment as previously stated.

Regards,

Denis Felikson and co-authors

Editor

Dear Denis Felikson and co-authors,

thank you for submitting to TC/TCD. You are certainly aware that papers accepted for TCD will appear immediately on the website and are open for comments and reviews. Before publication, submissions undergo a rapid access review by the editor to ensure initial quality standards. This quality control is not meant as a full scientific review but to ensure accordance to the journal remit as well as compliance with the general review criteria on originality, scientific quality and significance. The criteria for this evaluation can be found at: https://www.the-cryosphere.net/peer_review/review_criteria.html. Grades are from 1-4 (excellent - poor).

1. ORIGINALITY (Novelty): 2
The authors present Greenland ice sheet simulations using Bayesian calibration variants using different target observations. They show that the initial choice on the observational targets has important consequences for the committed global mean sea-level (GMSL) contribution of the Greenland Ice Sheet in 2100. Three observation map products are used: velocity changes, dynamics thickness changes or mass changes. In separate calibrations, they show that the posteriori GMSL uncertainties can significantly be reduced both for low- and high-end projections members. Moreover, important difference in maximum likelihood posteriori mass change are seen between the three observation types. The authors therefore suggest that more care has to be taken to better understand implications in the calibration procedure for future projections of ice-sheet evolution.

2. SCIENTIFIC QUALITY (Rigour): 3
The objectives of this work are well articulated and the methodology is adequately explained. While reading your manuscript, I had some questions which I already want to raise:

TERMINOLOGY
Throughout the manuscript you speak about the 'dynamic thickness change'. After digging into your manuscript, I think that I understand what you mean. Yet I am not sure if this is the correct term. I would rather speak about flux divergence. Please verify your terminology in the literature.

Author response: The term "dynamic thickness change" is used in literature, with "dynamic thinning" being similar, commonly used term to describe the recent dynamic changes that Greenland outlet glaciers are experiencing, which has predominantly been thinning and not thickening (e.g., Thomas et al., 2003; Pritchard et al., 2009; Flament et al., 2012; Khan et al., 2013; Wouters et al., 2015). Here, we choose to use dynamic thickness change instead of dynamic thinning because it can more generally refer to both dynamic thinning and thickening. However, we will change "dynamic thickness change" to "dynamic ice thickness change" to be more specific with our terminology, and we will add a sentence to explain that dynamic ice thickness change is equivalent to "flux divergence," as suggested by the editor.

INITIALISATION
As I understand your modelling setup, your starting point is an inversion of ice velocities in 2007 based on a given ice geometry. Could you add a few sentences on the initial modelled mass change over the observational period. Do you see any drift? Can you add a few words in the Methods and Discussions on this issue. I simply raise this point because I am worried that the analysis could be compromised by this initialisation variant.

Author response: There is, indeed, model drift and it is very difficult to distinguish between erroneous drift (caused by inconsistencies in the model initialization) and real drift (caused by the real Greenland Ice Sheet being out of balance at the initialization time). By comparing the modeled transient against the

Editor

observed transient over the calibration time period, 2007-2015, the goal of the Bayesian calibration is to assign higher weight to the model simulations that capture the observed transient response correctly. We will address this comment by adding a couple of sentences to the methods section to discuss the presence of model drift and the goal of Bayesian calibration in the face of this drift. We will also direct the reader to Nias et al. (2023), which has more discussion on model drift.

DYNAMIC THICKNESS CHANGE
- As I understand you, you infer the dynamic thickness change as the difference between the observed elevation change (dHdt) and the RACMO surface mass balance (SMB) corrected by a firn densification module. I wonder what the use of the firn densification module is. I suppose is that you have spatial information on the near-surface density. So it determines your conversion factor to water equivalents. Is that right? Moreover, I am not sure how you compute the simulated values for the dynamics thickness change. You certainly start off from the modelled elevation changes. Yet what is unclear to me is if you also correct with the firn densification module. If yes, I fear that you might introduce a bias. In any case, please make this clear because it might explain why this observation leads to the different calibration results.

Author response: We apply the firn densification correction only to the observations of surface elevation change and not the modeled surface elevation change. Firn densification can cause a change in surface elevation that is independent of the change caused by ice dynamics. For modeled ice dynamic thickness change, we remove the mass added or removed due to surface mass balance (SMB) but we do not account for firn densification because the model does not simulate firn. Because the model does not track changes in density, we do not need to account for firn densification when we calculate modeled ice dynamic thickness change. Thus, firn densification change is accounted for once (in the observations), without "double counting," and does not introduce a bias. A more detailed explanation follows.

To obtain observations of dynamic ice thickness change, we use estimates of surface elevation change due to (1) surface mass balance and (2) firn densification and remove these from the total observed surface elevation change, in order to quantify the dynamic ice thickness change. A detailed explanation of the calculation can be found in Csatho et al. (2014); Equation S1 in their supplementary provides the details of the calculation, summarized here:

$$H = h_{SMB} + h_{firn} + h_b + h_d$$

where $H$ is the total observed surface elevation change, $h_{SMB}$ is the surface elevation change due to surface mass balance (SMB), $h_{firn}$ is the surface elevation change due to firn densification, $h_b$ is the surface elevation change due to vertical crustal motion (i.e., glacial isostatic adjustment and elastic rebound), and $h_d$ is the surface elevation change due to dynamic ice thickness changes (i.e., flux divergence). In this paper, we assume $h_b$ to be negligible, and we account for $h_{SMB} + h_{firn}$ using the Institute for Marine and Atmospheric research Utrecht (IMAU) Firn Densification Model (FDM), which simulates total thickness change of the firn column caused by accumulation and ablation at the surface (i.e., the surface mass balance), as well as changes in firn thickness due to compaction. The IMAU-FDM models the change in firn thickness using an assumed near-surface density. With these signals removed, we are left with an estimate of the observed thickness change of the solid ice, due to dynamic changes in ice flow only (i.e., flux divergence).

To obtain modeled dynamic ice thickness change, we remove only the SMB component that is used as forcing for the model ($h_{SMB}$ in the equation above). We do not account for firn when we calculate the modeled dynamic ice thickness change because the ice flow model does not simulate firn – we use the SMB forcing (provided in units of water equivalent thickness) to add or remove ice from the surface by

changing the thickness at ice density. In other words, the ice flow model tracks the mass of the ice at each mesh node, with the thickness being the conversion of mass to thickness using ice density.

- Moreover, in my understanding there is no difference between using your dynamic thickness change values for calibration or directly using the density-corrected thickness changes. The reason is that you use a single SMB model. I might be wrong but please convince me otherwise. If I was not wrong, I sense that your methods can be formulated much more concisely avoiding the concept of 'dynamic thickness changes'.

Author response: As we describe above, our calculation of dynamic ice thickness change is used to estimate thickness changes due to changes in ice flow (i.e., changes in flux divergence), by removing thickness changes caused by changes in firn air content and accumulation and ablation at the surface (i.e., surface mass balance). Changes in firn air content do not produce mass changes but surface mass balance can produce mass change. Thus, to obtain density-corrected thickness changes that could be converted directly to mass change, we would need to add the surface mass balance to our dynamic ice thickness changes (this can be converted to units of mass via ice density). The reason that we have chosen to use dynamic ice thickness change, instead of a density-corrected thickness change is that dynamic ice thickness change is a more direct measure of the numerical ice sheet model than the density-corrected thickness change, which combines dynamic ice thickness change with firn thickness change.

- I further suggest that you integrate the density-converted elevation changes over entire Greenland and compare the resultant value to the ice-sheet-wide GRACE mass change. Optionally you could do this on a drainage basin level as done for the pre-processing for the calibration. Such a comparison could give a hint on the reason behind the different results after calibration.

Author response: This is an excellent idea, and one that we are planning to pursue in a follow-on paper. Reconciling observations of density-converted elevation changes to mass changes raises many open questions that are the focus of current on-going research. The open questions include not only the choice of surface mass balance and firn densification estimates used to calculate the density but also: (1) the method for interpolating the elevation change observations in time and space, (2) accounting for mass changes of peripheral glaciers and ice caps that are sampled by the mass change observations but not by the elevation change observations. In a follow-on paper, we would like to thoroughly address this topic and investigate the impact on the Bayesian calibration of ice sheet projections and, therefore, we would like to leave it out of the current paper.

MULTI-VARIATE CALIBRATION
Could you please elaborate a bit more on the inherent difficulties of a multi-variate calibration (all three observations) in a Bayesian framework. From the likelihood score, I do not see why you could not use multiple criteria. As Aschwanden and Brinkerhoff (2022) suggested a two-step procedure to circumvent this issue, I clearly miss something and I might not be alone.

Author response: This is an excellent suggestion, and we will add discussion on the difficulties of multi-variate calibration to the manuscript. Aschwanden and Brinkerhoff (2022) offer a very good approach to using observations of velocity to calibrate certain model parameters and mass change to calibrate other parameters and forcings. However, there are certain parameters that may benefit from simultaneous calibration from multiple observations. Our formulation of the likelihood score can easily incorporate multiple sources of model-observation residuals, however one challenge is in estimating the structural model uncertainty appropriately for each target variable. Another challenge is to account for structural deficiencies in, not only the model, but in the observations themselves. For example, the random component of uncertainty in the velocity change observations may be small but the data gaps introduce structural uncertainty in this observation type. This must be accounted for consistently across all

Editor

observations used in a multi-variate calibration. We will add a discussion of these challenges to the manuscript, including an acknowledgement of the Aschwanden and Brinkerhoff approach.

3. SIGNIFICANCE (Impact): 3
First and foremost, I fully appreciate the authors effort to forward our understanding on how to best initialise ice-sheet models with available observations while keeping a systematised framework on assessing model uncertainties. This is certainly a pressing task. In this sense, the article is of high relevance to the glaciological community. The article mainly promotes a Bayesian calibration framework and tries to raise awareness on the choice of the observational target. However, it does not give a final and robust advise. Additionally considering my concern on the scientific quality, I somehow have to moderate my assessment on the article significance.

4. PRESENTATION QUALITY: 1
The paper is well written and the structure is easy to follow. Findings are well supported by useful figures of mostly good quality.

Finally, please consider that the identified points are certainly not exhaustive. Yet they might well be indicative for issues that will potentially be picked up by reviewers. Moreover, find a list of more technical comments below. Please consider addressing all my comments at this stage. Alternatively address them during the revision process.

The editor, Johannes Fürst

##########################################################################################

**Technical comments:**
Title: I would refer to Greenland in the title to make it clear from the start that this study is not about a marine ice sheet.

Author response: We agree, and we will make the suggested change to the title.

Author response: We have revised the title to "Choice of observation type affects Bayesian calibration of Greenland Ice Sheet model simulations," adding a reference to Greenland, as suggested by the editor, and removing the word "projections," as suggested by reviewer #2 below.

P1L1: What do you mean by 'improving' uncertainties?

Author response: Here, we mean "advancing our understanding of the uncertainties" and we will revise the wording.

P1L5: What is a dynamic thickness change?

Author response: What is meant is the "dynamic ice thickness change." We think that adding the word "ice" will make this clearer, without being overly verbose in the abstract. Additionally, we will clarify in the main text that this is equivalent to change in flux divergence.

P2L22: score —> score to

Editor

Author response: We will make the suggested change.

P2L22: simulations —> simulation

Author response: We will make the suggested change.

P2L47-49: What is Section 3 about. Please specify.

Author response: We will add wording to describe Section 3 here.

P4L103-105: I would shift the description of the coordinate reference systems to Section 3.1

Author response: We will make the suggested change.

P4L111: Are these values in thickness change given for the full time period or as annual values.

Author response: These values are given for the full time period.

Fig. 1: Moderate quality of Figure. Is it possible to enlarge the different panels by re-ordering them (2x3 vs. 3x2). The sub-panel titles are very brief and confusing. Please add panel labels (a, b, c, …). What are the axis numbers?

Author response: We will make all of the recommended changes to this figure to improve the quality.

References:
Aschwanden, A., & Brinkerhoff, D. J. (2022). Calibrated Mass Loss Predictions for the Greenland Ice Sheet. *Geophysical Research Letters*, *49*(19), e2022GL099058.

Csatho, B. M., Schenk, A. F., van der Veen, C. J., Babonis, G., Duncan, K., Rezvanbehbahani, S., ... & Van Angelen, J. H. (2014). Laser altimetry reveals complex pattern of Greenland Ice Sheet dynamics. *Proceedings of the National Academy of Sciences*, *111*(52), 18478-18483.

Flament, T., Rémy, F., 2012. Dynamic thinning of Antarctic glaciers from along-track repeat radar altimetry. *Journal of Glaciology 58*, 830–840. https://doi.org/10.3189/2012JoG11J118

Khan, S.A., Kjær, K.H., Korsgaard, N.J., Wahr, J., Joughin, I.R., Timm, L.H., Bamber, J.L., van den Broeke, M.R., Stearns, L.A., Hamilton, G.S., Csatho, B.M., Nielsen, K., Hurkmans, R., Babonis, G., 2013. Recurring dynamically induced thinning during 1985 to 2010 on Upernavik Isstrøm, West Greenland. *Journal of Geophysical Research: Earth Surface 118*, 111–121. https://doi.org/10.1029/2012JF002481

Pritchard, H.D., Arthern, R.J., Vaughan, D.G., Edwards, L.A., 2009. Extensive dynamic thinning on the margins of the Greenland and Antarctic ice sheets. *Nature 461*, 971–975. https://doi.org/10.1038/nature08471

Thomas, R.H., Abdalati, W., Frederick, E., Krabill, W., Manizade, S.S., Steffen, K., 2003. Investigation of surface melting and dynamic thinning on Jakobshavn Isbræ, Greenland. *Journal of Glaciology 49*.

Editor

Wouters, B., Martín-Español, A., Helm, V., Flament, T., van Wessem, J.M., Ligtenberg, S.R.M., van den Broeke, M.R., Bamber, J.L., 2015. Dynamic thinning of glaciers on the Southern Antarctic Peninsula. *Science 348*, 899–903. https://doi.org/10.1126/science.aaa5727

Reviewer 1

**Review Felikson et al. (2022)**
This study aims to constrain and improve projections of the Greenland ice sheet evolution by making use of changes in three different observed quantities (mass changes, velocity changes and dynamic thickness changes) that are a measure the Greenland ice sheet dynamics. So far, Bayesian calibration of Greenland ice sheet model projections have been based on a single observation type.

I think the study presents a very useful framework to improve ice sheet model projections based on observations. However, as also stated in the study, using three different observation types do not give a clear answer on what is the best observation type to constrain the ice sheet model behavior and there might be a better metric when different observation types are combined. Therefore I am wondering whether the study as presented in the present form is convincing enough that model projections have been improved by using Bayesian calibration using different observation types. In addition, I would like to propose some minor clarifications on the methodological choices that you can read below.

Author response: We greatly appreciate the reviewer's comments, and we agree that more work remains to be done to incorporate information from several observation types and, ultimately, improve the methodology of Bayesian calibration for ice sheet projections. In this present study, our goals are (1) to show that Bayesian calibration using any of the observation types reduces uncertainty in the projections and (2) to analyze the discrepancies in the resulting posterior distributions from calibrations that use different observation types. We show that model projections of ice sheet contribution to sea level are improved because the spread in the prior distribution of projections is reduced by using any of the observation types for calibration. Thus, using any one of the observation types improves the uncertainty (spread) in sea-level change projections. However, we also show that care must be taken when using the maximum a posteriori estimate of sea-level change or the cumulative probability of higher-end sea-level change because the specific choice of observation type may bias these. We leave the reconciliation between the three observation types to future studies, as there is a significant amount of analysis to be done to investigate how to combine the information from all three types, as well as to investigate the use of derived metrics (e.g., spatial extent of thinning) and other sources of observations (e.g., outlet glacier terminus positions). For these reasons, we feel that our current paper is convincing and leaves room for future studies to build on our results. We appreciate all feedback and we have addressed all comments below.

**Main comments**
You explain that you coarsen the resolution in order to remove the spatial correlation between the different observations. However, if you look at Figure 1, there is still a strong correlation between the observed velocity change and the dynamic thickness change, especially for the fast outlet glaciers. It looks like it is a workaround to justify the neglection of a covariance matrix. I am also wondering if you are not losing too much important, detailed information by coarsening the resolution only to justify that the different variables are not correlated. Especially velocity changes at 50 km resolution remove a lot of detail.

Author response: As noted by the reviewer, our approach for coarsening the observations allows us to assume that the residuals are uncorrelated. The method that we use is consistent with previous studies that have used Bayesian calibration for ice sheet ensembles (e.g., Edwards et al., 2014; Ruckert et al., 2017; Nias et al., 2019; Brinkerhoff et al., 2021). In this way, our paper provides an evaluation of existing methods, which have been used in the literature. Using a covariance matrix would, in fact, allow more information and detail in the observations be used in the calibration. This would be a good direction for future work, but we leave it out of the scope of our paper in order to focus on evaluating calibration methods that have been used in past work.

Reviewer 1

Did you investigate the posterior probability distribution for higher spatial resolutions?

Author response: Early on in our analysis, we experimented with higher spatial resolutions for averaging the dynamic thickness change and velocity change. However, we ultimately chose to use a 50 km grid to alleviate the complication of properly accounting for spatial correlation in the observational uncertainties. As noted above, this could be improved upon but it is consistent with previous literature.

The model is initialized to the year 2007 and observations are used until 2015. What is the influence of the length of the observation time on the results? Could it be useful to use a longer observational time series, for instance by initializing the model to the year 2000 to increase the observational length?

Author response: The impact of the calibration time span would indeed be useful to investigate. However, our method for initializing the model, described in Nias et al. (2023), prevents us from initializing before 2007 because this is the earliest year for which spatially complete surface elevation and velocity datasets are available. Our simulations are initialized using a data assimilation method to invert for the basal friction coefficient and this requires an ice-sheet-wide dataset for ice sheet surface elevation. The 2007 surface elevation mosaic (Howat et al., 2014) is the highest quality, spatially complete, observation-based map of Greenland Ice Sheet surface topography that is dated furthest back in time. An older datasets of Greenland surface topography exists (Bamber et al. (2001) dataset that was used in the SeaRISE ice modeling project (Bindschadler et al., 2013), and was a tremendous asset in ice-sheet-wide modeling at the time. However, this dataset is based on radar altimetry, which has high uncertainty around the ice sheet margin. The aerial photogrammetry digital elevation model (Korsgaard et al., 2016) provides surface topography of the margin with low uncertainty but is not spatially complete, lacking elevation observations in the interior of the ice sheet. There is on-going work to compile a surface elevation dataset further back in time and, once these new datasets become available, it would be valuable to investigate the influence of the length of the observational timespan on the calibration. We discuss this briefly on line 324 and we leave this for future work.

You refer in the methodology to the Nias et al. paper in review, but there is no preprint available (or no reference in the reference list) so it is impossible to acquire the right information. For instance, the model ensemble details can be found in Nias et al. (in review). It would be interesting to have them in this manuscript as well. Also, you mention on L153 that the modeled mass change is aggregated within the same drainage basin as used for aggregating the observed mass change. Can you explain what that means, why and how that is done?

Author response: Nias et al. (in review) was not publicly available as a preprint at the time of our submission and we were instructed by the publisher to provide it to the journal editor, who could distribute it to reviewers upon request. Since the time of our submission, that paper now been published as Nias et al. (2023), providing complete details about the model setup and experimental design for the ensemble. We provide a summary of the ensemble in Section 2.1 and we will update the Nias et al. (in review) reference to Nias et al. (2023).

We will provide some additional explanation for the motivation for aggregating mass change within drainage basins and, as suggested by Reviewer 2, we will add the drainage basin outlines to Figure 1.

Author response: We have updated all references from Nias et al. (in review) to Nias et al. (2023) throughout the manuscript. We have also added text on lines 149-154 to provide more detail on why and how the aggregation within basins was done, and we have updated Figure 1 to show the observed mass changes at drainage-basin scale.

The goal is to narrow the committed Greenland ice sheet change projections. However, you are changing

Reviewer 1

not only model parameters but also the SMB, which you define as a forcing. Can you still speak of committed sea-level change if you change the SMB forcing?

Author response: We vary SMB forcing as part of the ensemble in order to account for uncertainty within the present-day SMB over the ice sheet. This way, our projections represent Greenland's committed sea-level contribution, given the uncertainty within the estimated present-day SMB, as well as the uncertainties within model parameters. This is explained in greater detail in Nias et al. (2023).

It would be nice to add some figures about the modelled Greenland ice sheet state. You show only 4 figures that basically focus only on the posterior probability distribution of committed GMSL rise due to melting of the Greenland ice sheet.

Author response: This is a very good suggestion, and we will add figures to show the modeled ice sheet state with panels to show velocity, thickness, and mass change for the heightest and lowest weighted simulations, similar to Figure 1 in Nias et al. (2023).

Author response: We have added figures showing the modeled GrIS states in 2007, 2015, and 2100, which show modeled states of velocity and surface elevation for the highest- and lowest-weighted ensemble members as revealed by the mass change calibration performed in Nias et al. (2023) (Figs. S1 and S2).

**Minor comments**
L61: It could be useful to give the values for the basal friction factor, the ice viscosity offset and the SMB offset for the 137 forward runs in the supplementary information. Could you add a justification for the offsets chosen?

Author response: The values for the ranges of these offsets, as well as a thorough justification for those chosen offsets, is given in Nias et al. (2023), including a supplementary file with the exact values for each ensemble member.

L151: The modeled quantities are regridded to 50 km x50 km. What is the original resolution of the model results?

Author response: We will add the original resolutions of each observation type, as well as the model results, to the appropriate subsections in Section 2.

L173-L175: The model uncertainty is a linear function of the observational uncertainty and you assign different multiplication factors for the different calibration to match the peak in the posterior distribution. Could you elaborate a bit more on the choice of these values? Because it looks contradictory to what you say on L185, that the median and maximum a posteriori GMSL are far apart.

Author response: This is an excellent question, and we will make this clearer in the text. We select the multiplication factor, $k$, for each observation type such that the value for the normalized probability of the peak in the posterior (i.e., the y-axis value) is approximately equal for all three calibrations. We did not make it clear in the text that it is the probability, and not the GMSL value at the peak probability, that we are aligning by varying the multiplication factor, $k$.

L207: That sounds like a logical consequence of the larger changes along the margin of the ice sheet.

Reviewer 1

Author response: This is a fair point, and we will note in the text that this is expected because we see larger observed changes around the margin.

L252: You discuss the firn thickness change as a potential bias. What is the modelled firn thickness change? Could you show a figure? Also you report a potential bias of 10 cm per year in the ice sheet interior. What is this number based on?

Author response: As suggested by the reviewer, we will add a supplementary figure to show thickness changes due to firn densification.

The 10 cm/yr bias was meant to represent a worst-case potential estimate of a potential bias in the firn thickness change. Our intention was to quantify the impact of such a bias on the calibration results. However, based on comments from both reviewers, we are going to remove this paragraph from the manuscript, as well as the associated plot, for two reasons:

1. A more thorough analysis should be done to investigate the impact of potential biases in the dynamic thickness change observations on the calibration, including from sources other than firn modeling.
2. Reviewer 2 commented: "I get the impression that you do not believe in the thickness change calibration because it is very dependent on the firn thickness change that is hard to constrain." While potential errors in the firn thickness change estimates are source of error in the dynamic ice thickness change observations, there are other sources of error that contribute. By focusing only on the impact of a potential bias in firn densification, we are unfairly presenting the importance of this source of error.

Author response: In response to this reviewer comment, as well as a comment from Reviewer #2 below, we have addressed this comment by replacing this paragraph, which focused solely on potential errors in the firn thickness change estimates, with a more general paragraph that discusses all sources of potential error in the observations, including the firn thickness change. The revised paragraph appears on lines 287-305. Additionally, we have added new figures to show the modeled firn thickness change (Fig. S3).

**Typo's**
L124: and and

Author response: This typo will be fixed.

L274: Open questions?

Author response: This typo will be fixed.

L284: correct 'to on the choice'

Author response: This typo will be fixed.

**Figures**
Figure 1: Something went wrong with the labels: please add/modify the labels (a), (b), (c), (d), (e) and (f). Please also adapt the colormap labels to make them uniform with the indication of the variable and the units between brackets. I would also add the resolution for the different gridded observations.

Reviewer 1

Author response: Thank you for noting these. We will make all the suggested improvements to this figure.

Figure 3: I do not see the blue curve with the Gaussian approximation.

Author response: We will add the blue curve – this was left out of the figure accidentally.

References:
Bindschadler, R. A., Nowicki, S., Abe-Ouchi, A., Aschwanden, A., Choi, H., Fastook, J., ... & Wang, W. L. (2013). Ice-sheet model sensitivities to environmental forcing and their use in projecting future sea level (the SeaRISE project). *Journal of Glaciology*, *59*(214), 195-224.

Bamber, J. L., Ekholm, S., & Krabill, W. B. (2001). A new, high-resolution digital elevation model of Greenland fully validated with airborne laser altimeter data. *Journal of Geophysical Research: Solid Earth*, *106*(B4), 6733-6745.

Brinkerhoff D, Aschwanden A, Fahnestock M (2021). Constraining subglacial processes from surface velocity observations using surrogate-based Bayesian inference. Journal of Glaciology 67(263), 385–403. https://doi.org/10.1017/jog.2020.112

Edwards, T. L.; Fettweis, X.; Gagliardini, O.; Gillet-Chaulet, F.; Goelzer, H.; Gregory, J. M.; Hoffman, M.; Huybrechts, P.; Payne, A. J.; Perego, M.; Price, S.; Quiquet, A. and Ritz, C. (2014). Probabilistic parameterisation of the surface mass balance–elevation feedback in regional climate model simulations of the Greenland ice sheet. The Cryosphere, 8(1) pp. 181–194.

Howat, I. M., Negrete, A., & Smith, B. E. (2014). The Greenland Ice Mapping Project (GIMP) land classification and surface elevation data sets. *The Cryosphere*, *8*(4), 1509-1518.

Korsgaard, N. J., Nuth, C., Khan, S. A., Kjeldsen, K. K., Bjørk, A. A., Schomacker, A., & Kjær, K. H. (2016). Digital elevation model and orthophotographs of Greenland based on aerial photographs from 1978–1987. *Scientific Data*, *3*(1), 1-15.

Nias, I. J., Cornford, S. L., Edwards, T. L., Gourmelen, N., & Payne, A. J. (2019). Assessing uncertainty in the dynamical ice response to ocean warming in the Amundsen Sea Embayment, West Antarctica. *Geophysical Research Letters*, 46(20), 11253-11260.

Nias, I. J., Nowicki, S., Felikson, D., & Loomis, B. (2023). Modeling the Greenland Ice Sheet's Committed Contribution to Sea Level During the 21st Century. *Journal of Geophysical Research: Earth Surface*, *128*(2), e2022JF006914.

Ruckert, K. L., Shaffer, G., Pollard, D., Guan, Y., Wong, T. E., Forest, C. E., & Keller, K. (2017). Assessing the impact of retreat mechanisms in a simple Antarctic ice sheet model using Bayesian calibration. *PLoS One*, *12*(1), e0170052.

Reviewer 2

Felikson et al. perform Bayesian calibration using different types of observational datasets as priors for an ensemble of model runs of the Greenland ice sheet under constant present-day climate. In general, better understanding the effect of calibration on the posterior probabilities is a very valuable endeavor. The study presented in the paper sheds some light on that task by showing that using different datasets for calibration of the prior distribution influences the posterior distribution quite substantially. In general, I recommend publishing the paper in TC, however, some comments have been addressed before doing so.

**Entire manuscript:**

-The underlying model ensemble is called "projections" in many places which will likely cause a misinterpretation of their results. I had to read to the methods to actually understand that the experiments underlying the calibration model the evolution of the Greenland ice sheet under current climate and are not informed by future SSP scenarios. A less careful reader could think that the numbers presented are sea-level projections. Make sure to change your wording, e.g. by adding "commitment under current climate" or "under constant present-day climate". Also add a timeframe over which this commitment is calculated earlier than in the methods (2100 or later?). This includes changing the title, the abstract and checking the rest of the manuscript.

Author response: We agree with the reviewer's comments, and we will make the following changes:
- Change our wording from "projections" to "commitment under current climate."
- Added the simulation timeframe to the abstract and introduction.

**Abstract:**

- line 6-9: What do you mean with "maximum a posteriori ice sheet mass change"?

Author response: The maximum a posteriori is the ice sheet mass change at the maximum of the posterior distribution. We have reviewed how we present our results and, in response to this comment and other comments below, we have decided to remove all references to the "maximum a posteriori" and, instead, present only the percentiles ($5^{th}$, $50^{th}$, $95^{th}$), with the $50^{th}$ percentile being equivalent to the median.

- end of abstract: what do you propose as a way forward?

Author response: We will add the following sentence to the end of the abstract: "Looking ahead, we present ideas for ways to improve Bayesian calibration of ice sheet projections."

The sentence in the conclusions on lines 323 – 326 states what we propose as potential ways forward: "future work should explore additional choices, such as the method for specifying model structural uncertainty, the timespan over which the calibration is done, the use of time series of observations rather than a snapshot of change, and the use of additional metrics derived from these observations."

**Introduction:**

- line 23: a "likelihood" of what? And "update" to what? The explanation of Bayesian calibration is quite cryptic, reformulate.

Author response: We will add detail to better describe the Bayesian calibration process in this section, including specifying what the "likelihood" and "update" are referring to.

Reviewer 2

**Methods:**

- Data/Model output processing: More detail is required on how you handle cells at the margins in your processing and regridding since those show most changes and hence are probably most important for you calibration. For example, is the ice front retreat that you impose part of the "dynamic thinning signal" or is this removed from the signal? And how does this compare to the observational dataset? How is this for the other datasets?

Author response: We will address this by providing additional detail in the manuscript on how mass change at the margins is handled, both in the observations and the model output. The ice front retreat is not part of the "dynamic thinning signal," which is calculated only at model mesh vertices where ice exists throughout the entire calibration period (2007-2015). Thus, the model mesh vertices that contain ice in 2007 but no ice in 2015, due to terminus retreat, are not used in the dynamic thinning calculation. Correspondingly, for observations of dynamic thinning, all observations beyond the ice extent are masked out. In other words, like the modeled dynamic thinning, the observed dynamic thinning is only calculated where there is ice during the entire calibration period (2007-2015).

For velocity change, the same processing is used; that quantity is calculated only where there is ice at the start and end of the calibration period, for both the observations and model results. For modeled mass change, the processing takes into account the ice mass lost due to retreat (and replaced with ocean water mass). For observed mass change, there is a constraint built into each mascon to treat it as either land or ocean. In our processing, only the land mascons are used in the calibration. The land mascons contain mass change signals from both land and grounded ice but in Greenland, the grounded ice mass change greatly outweighs the land mass change. We refer the reviewer to the second paragraph of Section 2.1 in Loomis et al. (2021) for more detail about the processing of the observed mass change over the ice sheet.

Author response: We have addressed this by providing additional detail on how modeled and observed changes at the ice sheet margin, where ice has retreated (lines 168-171). In our previous response above, we had erroneously stated that our calculation of modeled mass change takes into account ice mass lost due to retreat (and replacement with ocean water mass). Our processing does not account for this – we calculate mass change only at model grid points where ice remains in 2015. However, this unaccounted-for mass change due to ice retreat is negligible. As a conservative estimate, if 100 glaciers around the GrIS (each having 5 km width and 600 m height) retreat 3 km over 2007-2015, the mass difference between the lost ice and backfilled ocean water is about 40 Gt or about 2% of the total observed mass change from satellite gravimetry.

- line 134-136: apparently, mass changes are aggregated at a basin-scale, but figure 1 does not show this. You should update Figure 1 to show the basin-scale values you actually used, otherwise this is confusing. That you use basin-scale makes your results for mass changes comparable to other studies mentioned in the introduction that usually use aggregated values of mass change – extend on this in your discussion. If I misunderstood this comment here, and you did not do the calibration using the aggregated values, I suggest you to do this as this is a commonly used methodology and it would be interesting to compare with this.

Author response: This is correct – mass changes were aggregated at the basin scale. We will modify how we present observed mass changes in Figure 1 to show the changes at the basin scale, instead of within individual mascons.

- lines 155 and following: definitely more detail is required on the Bayesian calibration, implicit assumptions you make and how it is applied here. This methods is maybe not clear to all readers from The Cryosphere and this paper should hence make a better effort to introduce the methodology to their reader.

Reviewer 2

Moreover a reference to a standard book that described the methodology should be given. Lines 156-162 need clarification, i.e., that m stands EITHER for model parameters, forgings OR mass change in 2100, and just one of them, and that m does not lump all these together. Lines 163 and following: what is the reasoning behind using these terms to calculate the likelihood terms? What are the underlying assumptions, e.g., on the distribution of the priors? A definition of sigma_{o,i} is missing (is this based on the uncertainties shown in Fig. 1?). Furthermore, you should give the exact numerical form of the equations, for example in an appendix, that you use to calculate the respective terms used, i.e., the sigma, the likelihood, the posterior distribution (do you fit this in Figs 2 and 3 to the histogram resulting from weighting the prior histogram? Which method is used for fitting the curve?).

Author response: We will review our description of our Bayesian calibration method and we will provide more detail to make it clearer to the reader. Specifically:
1. We will search for a suitable textbook reference that describes the general principles of Bayesian inference
2. We will clarify our use of the symbol "m" and will introduce other symbols, where necessary, to distinguish the various quantities that "m" represent
3. We will add text to specify that our formulation for the likelihood score, and the terms within, is a typical formulation that has been used in previous literature (e.g., Edwards et al., 2014; Ruckert et al., 2017; Nias et al., 2019; Brinkerhoff et al., 2021)
4. We will clarify in Section 2.1 that the uncertainties that are used for (1) basal friction, (2) ice viscosity, and (3) surface mass balance forcing specify the assumed prior distributions on each of these parameters and forcings
5. We will clarify that sigma_{o,i} is what is displayed in Fig. 1
6. We will add details about how the posterior distribution curve was fit using the likelihood weights

Author response: We have revised our description of the Bayesian calibration method, as we stated in our previous response above. This included cleaning up the symbology, as well as referencing a paper (Kennedy and O'Hagan, 2001), rather than a textbook, as the seminal reference on the topic. The revised text appears on lines 177-203.

- Line 173: Does this manual adjustment of k influence anything else?

Author response: Yes, the choice of k will influence the width of the posterior, in addition to the peak of the normalized posterior probability distributions. We will add this to this part of the method section.

**Results:**

-Table 2: Add to the description that the "Prior" is of course not a "posterior".

Author response: We will clarify this in the table.

- Figure 4: Why are there "holes" in the residual plots for dv and dh? Description of RSS missing in legend. Do positive numbers mean that the ensemble member is faster / ticker than the observational dataset or is it the other way around? Why did you flip the colormap in the central pannels in comparison to the other panels (would be easier if they were similar)?

Author response: The "holes" are present where there are missing observations. The velocity change observations are missing data primarily in the southeast and we discuss as an issue that potentially contributes to the differences between the calibrations, in the paragraph starting on line 295. Similarly, the

dynamic thickness change observations have holes where laser altimetry measurements are missing, which can happen between satellite tracks and where there is a lack of airborne altimetry. Figure 7 in Schenk and Csatho (2012) shows a representative illustration of the gaps between altimetry measurements over the Greenland Ice Sheet.

We will add the description of "RSS" to the caption of Figure 4.

We describe the interpretation of the colors in the main text, but we will also add these to the caption for clarity. We tried to set up the colormaps so that, around the margin of the ice sheet where outlet glaciers are generally accelerating, dynamically thinning, and losing mass, blues indicate that the model is overestimating acceleration (i.e., there is too much modeled acceleration), overestimating dynamic thinning (i.e., there is too much modeled thinning), and overestimating mass loss (i.e., there is too much modeled mass loss). However, thanks to this reviewer comment, we have realized that we need to also flip the colormap for the last row showing mass loss. This would make the colors consistent in the way that we want. (Note that the text in the paragraph describing this figure is consistent with the figure in the submitted manuscript, so we will change the text accordingly.)

- line 185: "maximum a posterior sea-level" can be misunderstood to mean an upper bound on the sea-level contribution, not the value with maximum probability (if this is a correct assumption from my side?).

Author response: We have decided to remove the "maximum a posteriori" estimate of Greenland mass change from our results and discussion and, instead, will present and discuss the median ($50^{th}$ percentile), along with the $5^{th}$ and $95^{th}$ percentiles.

- lines 184-197: I wonder if these large differences between the calibration methods are also partly due to the fact that overall numbers are not very large. Or putting the question the other way around, would you expect similar large percentage differences when calibrating SSP5-8.5 projections instead of "committed mass loss"?

Author response: This is a good point, and it is certainly possible that the percentage differences will be smaller when calibrating an ensemble of the forced mass loss for SSP5-8.5 rather than the committed mass loss. We will add this as a caveat to the final discussion paragraph on lines 303-310.

- line 207-8: clarify sentence.

Author response: We will clarify this sentence.

- line 220: should it be "which overestimates thickness changes around Jacobshaven"?

Author response: We will change this wording to be "overestimates thinning" to make it clearer and more consistent with the previous sentence.

- line 224 and following: Not sure what you can learn from this exercise of comparing the RSS for these three simulations.

Author response: We are using the RSS of the residuals as a measure of the sensitivity of each modeled quantity to the different calibrations. For example, because the RSS of the velocity change residuals differs by 38% across all three calibrations (top row of Figure 4), modeled velocity is more sensitive to the chosen observation that is used for calibration than the modeled thickness change and mass change

Reviewer 2

(RSS residual differences of 16% and 18%, respectively). This indicates that thickness change and mass change are less sensitive to the chosen observation used for calibration.

**Discussion:**

- From your discussion I get the impression that you do not believe in the thickness change calibration because it is very dependent on the firn thickness change that is hard to constrain. If this is correct, I suggest that you make this one clear conclusion from your study, e.g., by suggesting to rather use the velocity or mass change calibration as these are less prone to these uncertainties.

Author response:  Uncertainty in firn thickness change estimates is an important source of error that should be investigated but it is too strong of a statement to say that we do not believe in the dynamic ice thickness change calibration due to this source of uncertainty alone. The dynamic ice thickness change observations are impacted by other sources of error: sparse sampling in space and time, as well as uncertainty in the surface mass balance. And the velocity and mass change observations are similarly impacted by errors that are specific to those observations. Our present study focuses on the combined impact of all error sources on the calibration, and we plan to investigate the sources of observational uncertainty more thoroughly, and their individual impacts on the calibration, for each of the three observation types in a future paper.

In response to this comment, as well as a comment from Reviewer 1 above, we will revise the paragraph on lines 252 – 261 by removing the additional calibration in which we introduce a 10 cm/yr bias to represent a potential error in the firn densification in the interior of the ice sheet. After discussing the reviewer comments, we feel that this unfairly focused too much attention on the possible errors in firn densification, without an equivalent discussion of other sources of error. We will replace this with a discussion of all of the potential sources of uncertainty.

Author response: In response to this reviewer comment, as well as a comment from Reviewer #1 above, we have addressed this comment by replacing this paragraph, which focused solely on potential errors in the firn thickness change estimates, with a more general paragraph that discusses all sources of potential error in the observations, including the firn thickness change. The revised paragraph appears on lines 287-305.

- line 273-278: You mention that open questions remain, but in the sentences following this, I cannot see any open questions that remain from Aschwanden and Brinkerhoff (2022). Is one open question whether it is better to use ice speed data or change in ice speed for calibration (why? Also the model initialisation in their study is different from the inversion used here, so maybe adding the direct velocity data information in their calibration is more like the step of using the velocity data for inversion done here)? Is it a bad that "the second step of calibration using mass change in Aschwanden and Brinkerhoff (2022) does not shift the posterior median estimate of ice sheet mass change" – because you make it sounds like this not a wanted result?

Author response: We agree that our statement that "open questions still remain" is not followed by a clear discussion of the questions that remain. We will revise this section to make it clear what questions are still unanswered in the existing literature, with more clarity on what Aschwanden and Brinkerhoff (2022) was able to answer. The reviewer makes a good point that the use of velocity observations in Aschwanden and Brinkerhoff (2022) is, in some ways, similar to initializing the model using velocity observations. We also do not mean to imply that a lack of a shift in the posterior median of ice sheet mass change, as seen in Aschwanden and Brinkerhoff (2022), is an unwanted result. We will clarify this point, as well.

Reviewer 2

Author response: We have addressed this comment, as well as the following comment, by revising our discussion of Aschwanden and Brikerhoff (2022) to present our results in the context of their work in a much clearer manner, removing the statement of open questions, highlighting that their use of velocity observations is akin to our model initialization as pointed out by the reviewer, and discussing ways to build upon their work and our work. The revised text appears on lines 314-321.

- line 278: Selling that you account for model uncertainty in contrast to this study is a bit too much, as you only use a very ad-hoc way of including it.

Author response: Our intent was not to imply that our study takes model uncertainty into account, and we will revise the wording in this section to make this more clear.

Using the terminology in Aschwanden et al. (2021), our ensemble samples parametric and aleatoric uncertainty but not model uncertainty. The reviewer makes a good point: our statement that "open questions still remain" (see comment above), followed by this referenced discussion of model uncertainty may lead readers to incorrectly interpret that our ensemble is an advancement over the Aschwanden and Brinkerhoff (2022) work. We will clarify this section accordingly.

- line 288-294: Not sure I understand what you want to imply with this part of the discussion.

Author response: We are glad that the reviewer brought this to our attention, and we will address this comment by revising the paragraph to make it more clear. The purpose of this part of our discussion is to highlight that the calibration method that we used, also used in previous studies, estimates structural model uncertainty by using a multiplier ($k$) to inflate the observational uncertainty, typically by an order or two orders of magnitude. With this approach, the spatial structure of the observational uncertainty (i.e., the relative magnitudes between data points) becomes more important than the absolute magnitude of the uncertainty in each individual observational data point. As the modeling community develops better methods for quantifying structural model uncertainty, we can move away from using this multiplier approach and towards having an estimate of structural model uncertainty that is independent of observational uncertainty. At that point, this argument will be moot. Currently, however, calibration that is used in the literature uses this approach and, here, we are drawing attention to this to emphasize to the observational communities that the current Bayesian calibration approaches are highly dependent on the relative errors between observation data points.

We will clarify this paragraph by doing the following:
- Revising the topic sentence to: "Current approaches to estimating structural model error in Bayesian calibration results in the spatial structure of observational uncertainty to be more important than the magnitudes of the observational uncertainty in each individual data point."
- Referencing the multiplicative factors used in previous studies.
- Ensure that we are using the wording of "observational uncertainty" and "structural model uncertainty" consistently in this paragraph and throughout the entire manuscript.

Author response: We have addressed this comment by making significant revisions, based on the reviewer's suggestions, that have clarified our meaning in this paragraph. After further discussion among the coauthors, we have simplified this paragraph to convey our intended message, which is that the spatial structure of observational uncertainty is important in the ad-hoc method that we employ, along with previous studies, to estimate structural model uncertainty. We have also added a sentence about the need to develop better understanding of structural model uncertainty that is independent of observational uncertainty. Finally, we have made our text more consistent in the use of "structural model uncertainty" throughout the entire manuscript. The revised text appears on lines 323-328.

Reviewer 2

- How much does adding the structural uncertainty in the ad-hoc manner (proportional to the observational uncertainty) affect your results, i.e., how would your results look like without accounting for this term?

Author response: This is a fair question. We will perform an additional set of calibrations, setting structural model uncertainty to zero to form the likelihood. We expect that neglecting structural model uncertainty will lead to the calibration applying high weight to a relatively smaller number of ensemble members, resulting in narrower posterior probability distributions for sea-level contribution. In other words, the calibration will "hone in" on a smaller number of ensemble members and will de-weight the rest. We will add this to the supplementary figures, with text added to the results and discussion in the manuscript.

Author response: We experimented with removing the structural model uncertainty term from our calibration (i.e., by setting the multiplier $k$ equal to zero) but this results in a set of calibration scores, $s\_j$, that are all equal to zero. Mathematically, this occurs because the denominator in Eqn. 2 becomes small, driving the exponential to zero. This stems from our use of a Gaussian likelihood function to construct Eqn 2. Our use of a Gaussian likelihood is consistent with previous literature (e.g., Nias et al., 2023) and using the same functional form for the likelihood for all observation types allows us to perform direct comparison of the results across all three calibrations, without introducing differences caused by using different likelihood functions. To further address this reviewer comment, we performed additional calibrations using lower values of the multiplier, $k$, to show the resulting posterior probability distributions (Figs. S5 and S6). A comprehensive analysis on the impact of various functional forms of the likehlihood on the calibration results should be performed but, because our study focuses on the differences in the calibrations caused by selecting different observation types, we leave this to future work. Our added text on this topic appears on lines 331-336.

**Conclusion:**

- line 315: I suggest rewording here, because it is not only the mass change calibration that leads to highest scoring members with undesired behavior, you found the same also for the other calibrations. I would rather write "As we show, using the mass change calibration – or any other single dataset for calibration - does not necessarily mean …"

Author response: We agree, and we will make the suggested change.

- line 318: "right answer" to what question?

Author response: We will remove this sentence. It is paraphrasing the previous sentence but, as the reviewer points out, it is unclear.

- line 328: you claim that you have shown that "utilizing different observation types in separate calibrations can yield additional insight into biases in the model ensemble", but the paragraph on that in the discussion (lines 261 and following) contains only once sentence on this ("For example, the highest-weighted ensemble member from the mass change ensemble overestimates acceleration (Fig. 4c) and underestimates dynamic thinning along almost the entirety of the GrIS margin (Fig. 4f).") and this appears to be more a problem of the observational dataset (the firn correction uncertainty) rather than a bias in the mode ensemble?

Author response: We will develop additional examples for how the different observation types can yield additional insights, and we will add this to the discussion. The specific example about the underestimate

of dynamic thinning along the GrIS margin does not necessarily correspond to a potential bias with the firn densification because most of the margin is an ablation zone, without firn. We will explicitly state this in this part of the discussion.

- What do you recommend, based on your results, to the modeling community to make meaningful model calibration in the future?

Author response: We have provided some guidance for future work on lines 323 – 330, but we will also add text to state that the modeling community should develop robust methods to quantify structural model uncertainty for velocity change, dynamic ice thickness change, and mass change, which could then be used to perform one multi-variate calibration using all three observation types simultaneously.

References:

Aschwanden, A., Bartholomaus, T. C., Brinkerhoff, D. J., & Truffer, M. (2021). Brief communication: A roadmap towards credible projections of ice sheet contribution to sea level. *The Cryosphere, 15*(12), 5705-5715.

Aschwanden, A., & Brinkerhoff, D. J. (2022). Calibrated Mass Loss Predictions for the Greenland Ice Sheet. *Geophysical Research Letters*, *49*(19), e2022GL099058.

Brinkerhoff D, Aschwanden A, Fahnestock M (2021). Constraining subglacial processes from surface velocity observations using surrogate-based Bayesian inference. Journal of Glaciology 67(263), 385–403. https://doi.org/10.1017/jog.2020.112

Edwards, T. L.; Fettweis, X.; Gagliardini, O.; Gillet-Chaulet, F.; Goelzer, H.; Gregory, J. M.; Hoffman, M.; Huybrechts, P.; Payne, A. J.; Perego, M.; Price, S.; Quiquet, A. and Ritz, C. (2014). Probabilistic parameterisation of the surface mass balance–elevation feedback in regional climate model simulations of the Greenland ice sheet. The Cryosphere, 8(1) pp. 181–194.

Loomis, B. D., Felikson, D., Sabaka, T. J., & Medley, B. (2021). High-Spatial-Resolution Mass Rates From GRACE and GRACE-FO: Global and Ice Sheet Analyses. *Journal of Geophysical Research: Solid Earth, 126*(12), e2021JB023024.

Nias, I. J., Cornford, S. L., Edwards, T. L., Gourmelen, N., & Payne, A. J. (2019). Assessing uncertainty in the dynamical ice response to ocean warming in the Amundsen Sea Embayment, West Antarctica. *Geophysical Research Letters*, 46(20), 11253-11260.

Ruckert, K. L., Shaffer, G., Pollard, D., Guan, Y., Wong, T. E., Forest, C. E., & Keller, K. (2017). Assessing the impact of retreat mechanisms in a simple Antarctic ice sheet model using Bayesian calibration. *PLoS One*, *12*(1), e0170052.

Schenk, T., & Csatho, B. (2012). A new methodology for detecting ice sheet surface elevation changes from laser altimetry data. IEEE *Transactions on Geoscience and Remote Sensing*, 50(9), 3302-3316.

---

## Author Response (AR2)

**Summary**

Thank you to the editor and anonymous reviewers. Your feedback throughout the review process has allowed us to improve the manuscript. With this revision, we submit the final technical corrections and provide responses to the reviewer comments below.

**Report #1**

Thanks to the authors for addressing the comments, the manuscript has much improved! And I think it is a great contribution to TC. I have only two comments left:

(1) Line 105: You mention that you subtract firn thickness changes from the surface elevation changes to obtain the dynamic thickness changes. But should not also a change in surface mass balance be accounted for? I thought that this is done by using RACMOv2.3p2 model results for the period from 207 to 2015

[Author response] To make it clearer that we account for both SMB and firn density changes, we have added this clause to the end of the sentence on line 105: "… which simulates thickness changes due to SMB."

(2) One question that seems a bit obvious: why you did not test combining the three types of observations for calibration?

[Author response] This was suggested in a previous revision, and we agree that the next logical step is to try to combine all three observation types into one calibration. However, in order to combine the three observations into one calibration, there are certain challenges that need to be addressed. The primary challenge is the need to better estimate the structural uncertainties in both the model and observations. This needs to be done consistently across all three observations observations used in a multi-variate calibration. In a previous revision, we addressed a reviewer comment on this same topic by adding discussion text to address the challenges that remain and the path forward.

**Report #2**

Remark on Figure 1:

Thanks you for adapting the figure and showing the aggregated mass change (uncertainty). I still find the figure caption not logically structured, please describe chronological panel a), b), c), d) e) and f).

[Author response] We have adjusted the caption for Figure 1 to introduce each of the panels in an alphabetical order: a), b), c), d), e), and f).